# Discovery of a potent HMG-CoA reductase degrader that eliminates statin-induced reductase accumulation and lowers cholesterol

Shi-You Jiang [1], Hui Li[2], Jing-Jie Tang[3], Jie Wang[2], Jie Luo[1], Bing Liu[2], Jin-Kai Wang[1], Xiong-Jie Shi[1], Hai-Wei Cui[2], Jie Tang[2], Fan Yang[2], Wei Qi[4], Wen-Wei Qiu[2] & Bao-Liang Song [1]

Statins are inhibitors of HMG-CoA reductase, the rate-limiting enzyme of cholesterol bio-synthesis, and have been clinically used to treat cardiovascular disease. However, a para-doxical increase of reductase protein following statin treatment may attenuate the effect and increase the side effects. Here we present a previously unexplored strategy to alleviate statin-induced reductase accumulation by inducing its degradation. Inspired by the observations that cholesterol intermediates trigger reductase degradation, we identify a potent degrader, namely Cmpd 81, through structure–activity relationship analysis of sterol analogs. Cmpd 81 stimulates ubiquitination and degradation of reductase in an Insig-dependent manner, thus dramatically reducing protein accumulation induced by various statins. Cmpd 81 can act alone or synergistically with statin to lower cholesterol and reduce atherosclerotic plaques in mice. Collectively, our work suggests that inducing reductase degradation by Cmpd 81 or similar chemicals alone or in combination with statin therapy can be a promising strategy for treating cardiovascular disease.

[1] Hubei Key Laboratory of Cell Homeostasis, College of Life Sciences, Institute for Advanced Studies, Wuhan University, 430072 Wuhan, China. [2] Shanghai Engineering Research Center of Molecular Therapeutics and New Drug Development, School of Chemistry and Molecular Engineering, East China Normal University, 200241 Shanghai, China. [3] State Key Laboratory of Molecular Biology, Institute of Biochemistry and Cell Biology, University of Chinese Academy of Sciences, Chinese Academy of Sciences, 200031 Shanghai, China. [4] School of Life Science and Technology, ShanghaiTech University, Shanghai 201210, China. These authors contributed equally: Shi-You Jiang, Hui Li, Jing-Jie Tang. Correspondence and requests for materials should be addressed to W.-W.Q. (email: wwqiu@chem.ecnu.edu.cn) or to B.-L.S. (email: blsong@whu.edu.cn)

Cardiovascular disease (CVD) is a public health crisis accounting for more than 30% of all deaths globally[1]. Elevated serum level of cholesterol, particularly low-density lipoprotein cholesterol (LDL-C), is recognized as one of the major risk factors for CVD[1]. Statins, owing to the efficacy in reducing blood LDL-C concentration[2–4], remain as the most widely prescribed medications for both primary and secondary CVD prevention[5–7].

Statins act by competitively binding to the catalytic domain of 3-hydroxy-3-methylglutaryl coenzyme A reductase (HMG-CoA reductase, HMGCR)[8,9] and blocking the conversion of HMG-CoA to mevalonate, the rate-limiting step of cholesterol bio-synthesis. Notably, sterols and nonsterol isoprenoids derived from mevalonate normally govern the protein level of HMGCR through a multivalent feedback mechanism[10]. Briefly, insufficient cholesterol production enhances transcription of the HMGCR gene through activating the sterol regulatory element (SRE)-binding protein (SREBP) pathway[11]. Meanwhile, attenuated generation of lanosterol or 24,25-dihydrolanosterol (24,25-DHL) and geranylgeraniol slows down the degradation of HMGCR protein, extending half-life of the enzyme[12–14]. These effects converge to increase the amount of HMGCR up to hundreds of folds, which even causes the morphological change of the endo-plasmic reticulum (ER) in cultured cells[15–17]. Observed in mice[18–20], rats[21,22], miniature pigs[23], and humans[24], similar compensatory increase of HMGCR can weaken the effectiveness of statins and promote a need of drug at higher doses that are often associated with increased risks of new-onset type 2 diabetes mellitus[25,26] and myalgias[27–29]. The compensatory HMGCR elevation may also worsen the outcome in certain patients who stop taking statins after long term use[30,31], as HMGCR has been induced to a very high level, releasing of the inhibitor statin may contribute to causing rebound effects. Although this drawback has been recognized since the clinical application of statin[15], there is no strategy to prevent statin-induced HMGCR increment.

The HMGCR protein consists of two domains: (1) the cyto-plasmic C-terminal domain conferring all of the catalytic activity of the enzyme[8,32] and (2) the membrane-bound domain that interacts with ER-anchored Insig-1 and Insig-2 when cellular sterol accumulates[33,34]. The Insig-associated gp78, a ubiquitin ligase localized in the ER, and other cofactors then catalyze the ubiquitination of HMGCR at lysine 89 and 248[14,35–37], leading to its degradation by proteasomes[38,39]. The physiological HMGCR degraders (not typical proteolysis-targeting chimaera degraders[40]) include oxygenated cholesterol derivatives such as 25-hydroxycholesterol (25-HC) and 27-hydroxycholesterol (27-HC)[41–44], as well as cholesterol synthesis intermediates such as lanosterol and 24,25-DHL[12,45]. While the oxysterols prove useful in studying the feedback regulation of cholesterol metabolism, the likelihoods of developing them into drugs are little because oxysterols increase hepatic fatty acid biosynthesis through bind-ing to liver X receptor (LXR) and activating expression of the SREBP-1c gene[46,47]. Lanosterol and 24,25-DHL, on the other hand, may enter the Bloch and/or Kandutsch-Russell pathway and be converted to cholesterol[48].

In this study, we identify a potent HMGCR degrader, Cmpd 81 (also named as HMG499), elucidate the mechanism of Cmpd 81 action in stimulating HMGCR degradation and assess its ther-apeutic efficacy, when used alone or synergistically with stain, for treatments of hyperlipidemia and atherosclerosis.

## Results

**Statins induce compensatory increases of HMGCR protein**. We first evaluated statin-induced HMGCR increase in a physiological setting. Mice were gavaged with vehicle or lovastatin for 7

consecutive days and hepatic HMGCR protein was examined using immunoblotting. The amounts of HMGCR protein in mice receiving lovastatin were 58-fold higher than that from vehicle-treated mice (Fig. 1a, b). On the other hand, the Hmgcr mRNA abundance in lovastatin-treated group was only 2.5-fold higher than control mice (Fig. 1c).

We next investigated the effect of statins on the level of HMGCR protein in cultured cells. As shown in Fig. 1d, e, a dose-dependent increase in HMGCR protein was observed following lovastatin treatment, with a maximum increment of ~15-fold (Fig. 1d, e). In parallel, the mRNA abundance of Hmgcr was increased only 3.6-fold in the presence of lovastatin up to 10 µM (Fig. 1e). Similarly, other clinically used statins including simvastatin, pravastatin, fluvastatin, atorvastatin and rosuvastatin dramatically increased the HMGCR protein by approximately 15-, 12-, 11-, 9-, and 17-fold, respectively (Supplementary Fig. 1a, c, e, g, i), while slightly upregulated the Hmgcr mRNA by 3.5-, 2.4-, 2.6-, 1.8-, and 2.3-fold (Supplementary Fig. 1b, d, f, h, j). Collectively, these results showed that induction of HMGCR protein accumulation was a common phenomenon after the treatment of statins.

When statins blocked cholesterol synthesis through inhibiting HMGCR, the amount of HMGCR was under two levels of regulations. On one hand, HMGCR protein was stabilized as the sterol-induced ubiquitination of HMGCR by its E3 ligase, such as gp78, was blocked[35,37,49]. On the other hand, SREBP was activated and the transcription of SREBP target genes in cholesterol biosynthesis (including HMGCR) and cholesterol uptake were upregulated and thereby HMGCR protein can be augmented[11] (Supplementary Fig. 2a, b). To distinguish the contributions of each regulation on the level of HMGCR protein, we examined lovastatin-induced HMGCR level in SRD-13A, a Scap-deficient CHO-7 cell in which SREBP pathway cannot be activated. As anticipated, no change was found in the amounts of Hmgcr mRNA, while the amounts of HMGCR protein were still increased by as much as 8-fold (Fig. 1f, g), suggesting that the HMGCR stabilization is the major reason for statin-induced HMGCR increment. Considering that HMGCR is degraded through the ubiquitination at lysine 89 and 248[14], we further evaluated the effect of lovastatin on the wild type (WT) and lysine-to-arginine mutated (K89R/248 R) HMGCR. The over-expressed WT HMGCR protein could be elevated by 6-fold with 1 µM of lovastatin. Although the ubiquitination-resistant K89R/K24R HMGCR exhibited a higher basal level, statin treatment did not induce the protein level further increase (Supplementary Fig. 2c, d). Notably, the mRNA levels of both WT and mutated Hmgcr did not show obvious change (Supplementary Fig. 2d). These data demonstrated that statins strongly increased HMGCR protein mainly through preventing the ubiquitination and degradation of HMGCR.

**Structural optimization for HMGCR degrader**. We next sought to establish a reporting system to conveniently monitor the degradation of HMGCR. It is known that the N-terminal trans-membrane domain of HMGCR senses sterol levels in the ER membrane and determines the half-life of the whole protein[34,35]. We thus generated a cell line stably expressing the transmem-brane domain of HMGCR fused with green fluorescent protein (GFP) as a reporter (HMGCR (TM1-8)-GFP) (Fig. 2a). We first validated the system in the presence of 1 µM lovastatin using cholesterol and 24,25-DHL, which is known to induce degrada-tion of HMGCR[12]. Indeed, 24,25-DHL stimulated degradation of endogenous and exogenous HMGCR was much more potent than cholesterol (Fig. 2b, c). Of note, ectopically expressed HMGCR (TM1-8)-GFP and endogenous HMGCR degraded to a similar extent following cholesterol or 24,25-DHL treatments (Fig. 2b, c).

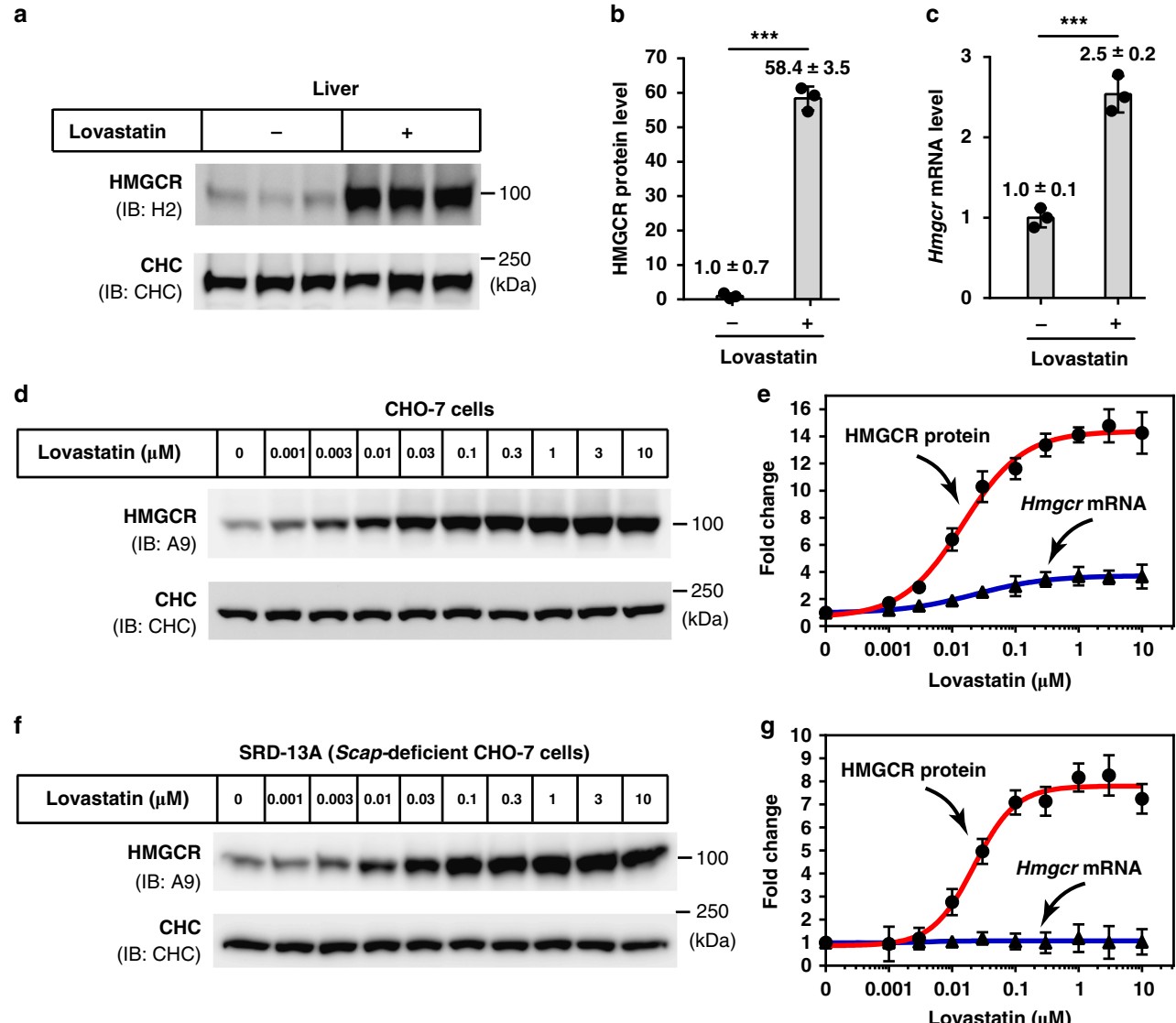

**Fig. 1** Lovastatin causes a substantial accumulation of HMGCR protein. **a–c** Male C57BL/6J mice ($n = 3$ per group) were gavaged with vehicle or lovastatin (60 mg/kg/day) once daily for 7 days. Livers were harvested for immunoblotting and RT-qPCR. **a** Immunoblotting analysis of HMGCR protein from membrane fractions. Clathrin heavy chain (CHC) was a loading control. **b** Quantifications of HMGCR protein shown in **a**. The HMGCR protein level of mice gavaged with vehicle was defined as 1. **c** Quantifications of *Hmgcr* mRNA level by RT-qPCR. The *Hmgcr* mRNA level of vehicle-treated mice was defined as 1. *Cyclophilin* was used as the reference gene. **d**, **e** CHO-7 cells were treated with indicated concentrations of lovastatin for 16 h, then harvested for immunoblotting and RT-qPCR. **d** Immunoblotting analysis of HMGCR protein. **e** Dose–response curves of HMGCR protein and *Hmgcr* mRNA levels. HMGCR protein and *Hmgcr* mRNA levels of DMSO-treated cells were defined as 1. *Glyceraldehyde-3-phosphate dehydrogenase* (*Gapdh*) was a reference gene. **f**, **g** SRD-13A cells were treated with lovastatin for 16 h. **f** HMGCR protein was analyzed by immunoblotting. **g** *Hmgcr* mRNA levels were measured by RT-qPCR. Data are from three independent experiments and presented as mean ± SD. ***$P < 0.001$, unpaired two-tailed Student's *t*-test. Source data are provided as a Source Data File. Uncropped immunoblots are shown in Supplementary Fig. 9

A dose-dependent decrease in GFP fluorescent signal was consistently observed in cells incubated with 24,25-DHL using the high-content imaging system (Fig. 2d). The half maximal effective concentration ($EC_{50}$) values, as determined by quantifications of GFP signal, were approximately 1.5 μM for 24,25-DHL and 30.4 μM for cholesterol (Fig. 2e). These results suggest that the HMGCR (TM1-8)-GFP reporting system can be used as a reliable assay to screen for compounds promoting HMGCR degradation.

Previous studies have shown that C4 methylation confers lanosterol or 24,25-DHL a greater potency than cholesterol in promoting HMGCR degradation[12]. To determine the optimal structural features of HMGCR degrader, we first synthesized a set of compounds based on the cholesterol scaffold with different modifications at the C4 position (Fig. 3a). Compound 7 (**7**), which harbors a 4α, 4β-dimethyl group on cholesterol, enhanced the potency from an $EC_{50}$ of 30.4 μM (Fig. 2e) to 14.6 μM in HMGCR (TM1-8)-GFP fluorescent assay (Fig. 3a). Substituting the dimethyl moiety to others, such as the 4α, 4β-diethyl group (**8**), 4α, 4β-diallyl group (**9**), 4-spirocyclohexyl group (**10**), 4-ethyl, 4-methoxyethyl group (**11**), or 4α, 4β-dimethoxyethyl group (**14**), did not decrease the HMGCR (TM1-8)-GFP intensity (Fig. 3a). Thus, a pair of methyl groups seems to be the best substitution at the C4 position of sterol facilitating HMGCR degradation.

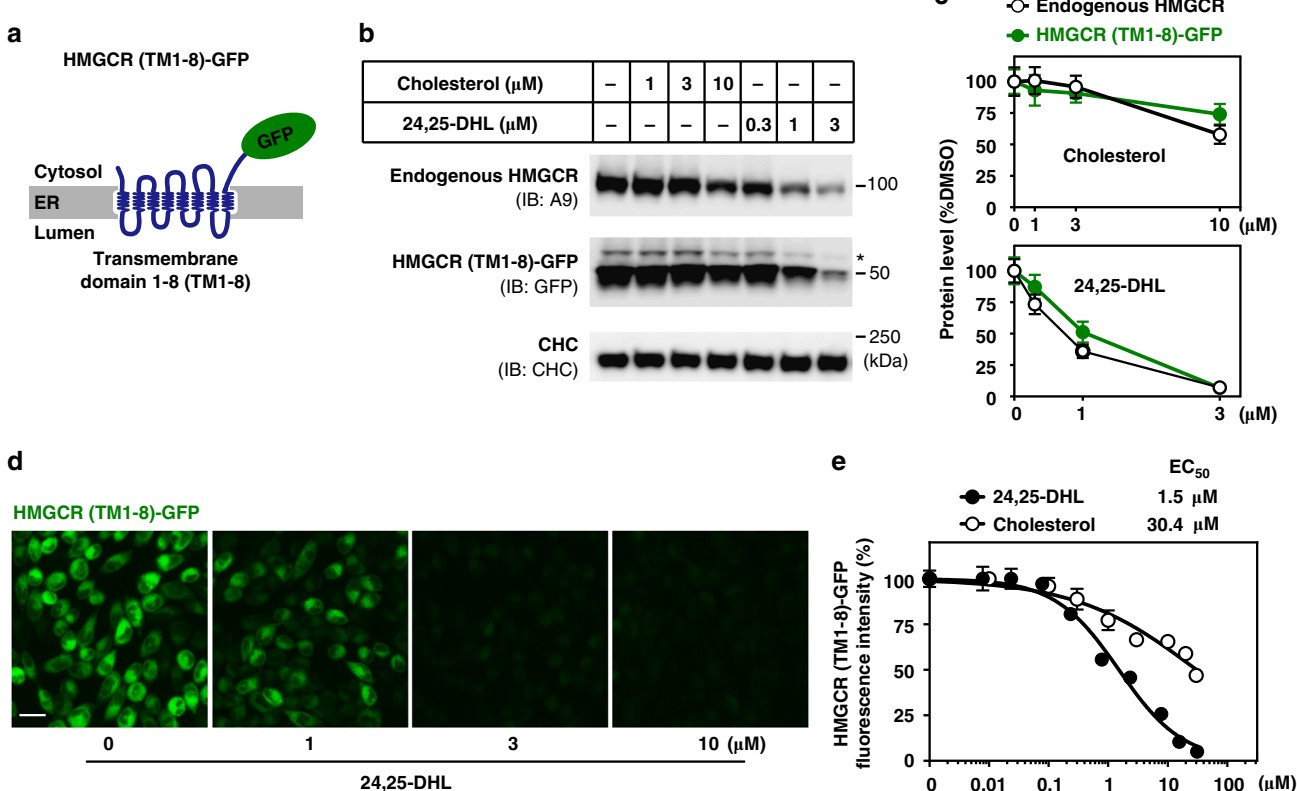

**Fig. 2** A reporting system that measures HMGCR degradation. **a** Schematic of the HMGCR (TM1-8)-GFP fusion protein. TM, transmembrane. **b**, **c** CHO-7 cells stably expressing HMGCR (TM1-8)-GFP (CHG) were incubated with cholesterol or 24,25-DHL at indicated concentrations for 16 h and harvested for immunoblotting. **b** Endogenous HMGCR (IgG-A9) and overexpressed HMGCR (TM1-8)-GFP (polyclonal rabbit anti-GFP) protein were analyzed by immunoblotting. Asterisk represents a non-specific band. **c** Quantification of endogenous and overexpressed HMGCR protein levels in response to cholesterol or 24,25-DHL shown in **b**. The mean intensity of HMGCR protein bands in DMSO-treated cells was defined as 100. **d**, **e** CHG cells were incubated with varying concentrations of 24,25-DHL for 16 h. Cells were then fixed for immunofluorescence analysis. **d** Representative images showing a dose-dependent decrease of GFP signals following 24,25-DHL treatment. Scale bar, 20 μm. **e** Dose–response curves of HMGCR (TM1-8)-GFP fluorescent intensity to varying concentrations of cholesterol or 24,25-DHL. The GFP intensity of DMSO-treated cells was defined as 100. The mean $EC_{50}$ values of cholesterol and 24,25-DH were 30.4 μM and 1.5 μM, respectively. Data are from three independent experiments and presented as mean ± SD. Source data are provided as a Source Data File. Uncropped immunoblots are shown in Supplementary Fig. 9

We next set out to improve the efficacy of **7** by attaching different groups to the C7 position (Fig. 3b). Compound 35 (**35**), adding of a 7β-hydroxyl group based on the scaffold of **7**, accelerated HMGCR degradation with an $EC_{50}$ of 3.6 μM, improving the activity four times than **7** (Fig. 3a, b). The analog with a 7α-hydroxyl group (**36**) displayed a weaker activity than **35** (Fig. 3b). Replacing the hydroxyl group at the C7 position with an azide group (**18**), an amino group (**19**), an acetylamino group (**22**), a keto group (**33**) or an oxime group (**34**) completely or slightly counteracted the ability of **7** to trigger HMGCR degradation (Fig. 3a, b). Together, these results indicate that the 4α, 4β-dimethyl group and 7β-hydroxyl group synergistically enhance the ability of cholesterol to promote HMGCR degradation.

Furthermore, we explored optimal side chain modifications by changing the terminal moiety and the length of the carbon chain based on the skeleton of **35** (Fig. 4 and Supplementary Fig. 3). The derivatives with more than four carbons in the side chain generally exhibited stronger activity than those with only four, exhibiting smaller $EC_{50}$ values, suggestive of enhanced potency in degrading HMGCR. In particular, those harboring a branched terminal moiety, for examples, the 2-hydroxy-2-propyl group (**79**, **100**, **117**) and the 1-ethyl-1-hydroxypropyl group (**80**, **101**, **118**),

were even more effective in degrading HMGCR protein. Interestingly, the $EC_{50}$ value of Compound 81 (Cmpd 81, **81**) was as low as 0.41 μM when the four-carbon side chain was terminated by a 1,6-heptadien-4-hydroxy moiety. Thus, starting from cholesterol whose $EC_{50}$ is 30.4 μM (Fig. 2e), we have successfully obtained several sterol derivatives with 50-fold higher potency in eliminating HMGCR (TM1-8)-GFP (Fig. 4 and Supplementary Fig. 3).

**Cmpd 81 promotes ubiquitination and degradation of HMGCR.** We next measured the degradation of endogenous HMGCR to confirm the effect of the compounds. The sterol analogs with $EC_{50}$ values close to or less than 1 μM, as determined by the HMGCR (TM1-8)-GFP reporting system, were added into cell culture medium at varying concentrations. All these tested compounds (such as **79**, **80**, Cmpd 81, **87**, **88**, **100**, **101**, **108**, **109**, **117**, **118** and **124**) performed similar or even better in degrading endogenous HMGCR comparing with 24,25-DHL (Supplementary Fig. 4a, d). We chose to further examine Cmpd 81 because 1 μM of this sterol derivative was sufficient to cause a more than 80% reduction in HMGCR (Supplementary Fig. 4a). Consistent with HMGCR (TM1-8)-GFP reporter imaging assay, the

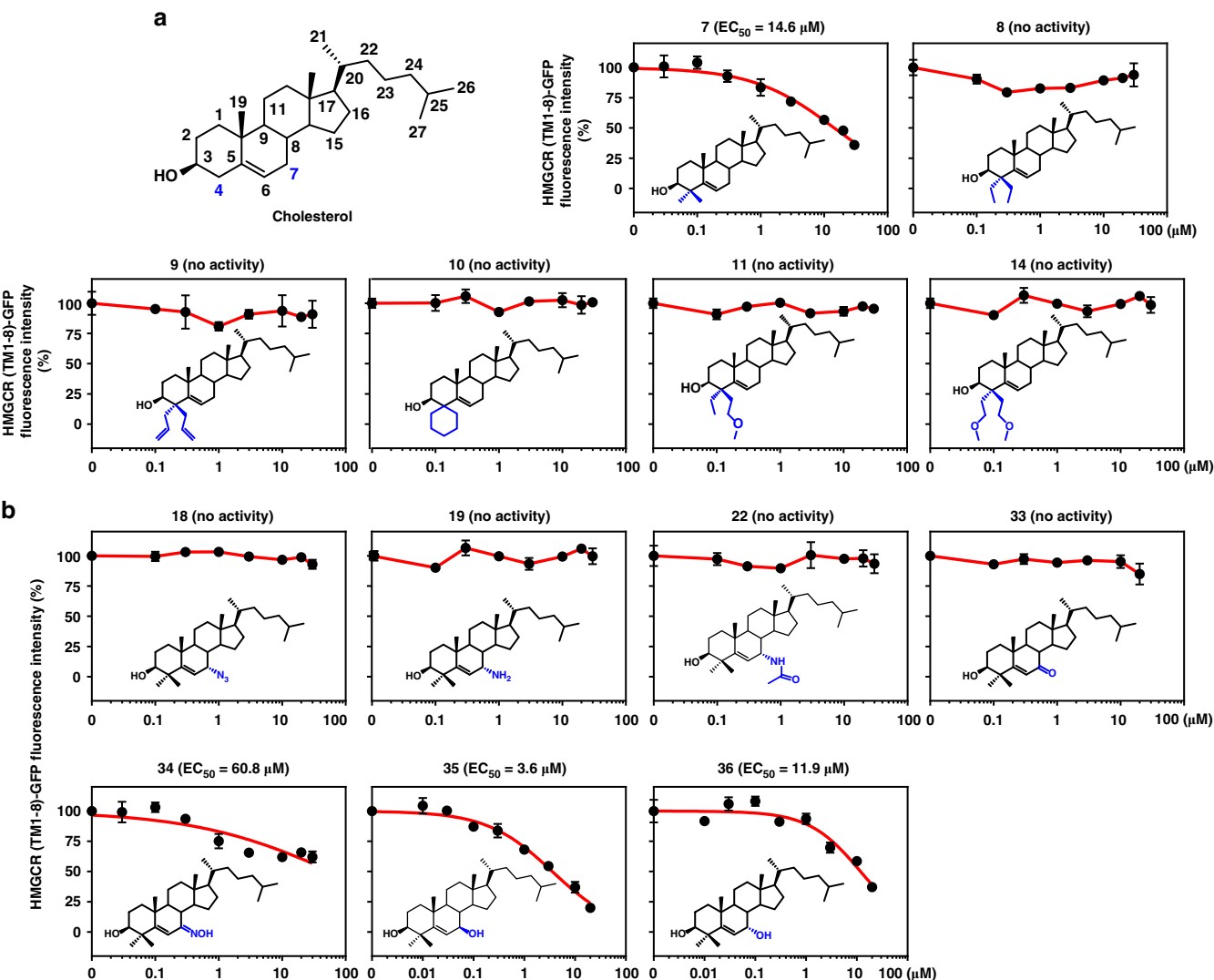

**Fig. 3** Identification of essential structural features of HMGCR degrader. **a**, **b** CHG cells were incubated with indicated compounds for 16 h. The GFP intensity of DMSO-treated cells was defined as 100. **a** Dose–response curves of HMGCR (TM1-8)-GFP fluorescent intensity to cholesterol analogs with different modifications at C-4 position. 4,4-Dimethyl cholesterol (**7**) was able to degrade HMGCR protein. **b** Dose–response curves of HMGCR (TM1-8)-GFP fluorescent intensity to 4,4-dimethyl cholesterol derived analogs with different moieties at C-7 position. 4,4-Dimethyl 7β-hydroxyl cholesterol (**35**) had improved activity of inducing HMGCR degradation. Data are from three independent experiments and presented as mean ± SD. Source data are provided as a Source Data File

dose–response immunoblotting analysis revealed an EC$_{50}$ value of 0.39 μM for Cmpd 81 to decrease endogenous HMGCR (Fig. 5a, b), approximately four times more potent than 24,25-DHL (Supplementary Fig. 4e, f). Importantly, unlike oxysterols[46,47], the above examined compounds (including Cmpd 81) did not induce the expression of *SREBP-1c*, *ABCA1* and *ABCG8* (Supplementary Fig. 5). These data suggest that Cmpd 81 and its analogs do not activate the LXR signaling pathway.

It is known that HMGCR is ubiquitinated and targeted to proteasome for degradation when cellular sterol level elevates[14,34–36,38,39]. We explored whether Cmpd 81 triggers HMGCR degradation by the ubiquitin–proteasome pathway. Indeed, addition of the proteasome inhibitor MG-132 completely blocked HMGCR degradation induced by Cmpd 81 (Fig. 5c). Concordantly, a dose-dependent increase in the ubiquitination of endogenous HMGCR was detected following Cmpd 81 treatment (Fig. 5d). In cells co-expressing wild-type HMGCR and Insig-1, Cmpd 81 also caused a markedly increase in the ubiquitination of

transfected HMGCR (Fig. 5e), accompanied by a reduction of the protein level (Fig. 5f). However, arginine replacement of lysine 89 and 248 blunted the effect of Cmpd 81 on HMGCR degradation (Fig. 5f). Notably, Cmpd 81-induced disappearance of endogenous HMGCR was completely abolished in SRD-15 cells lacking Insig-1 and Insig-2 proteins[16] (Fig. 5g). These findings suggest that Cmpd 81 promotes ubiquitination and degradation of HMGCR in a process similar to oxysterols and 24,25-DHL, which requires Insig proteins.

**Cmpd 81 lowers cholesterol and prevents atherosclerosis**. We further determined whether Cmpd 81 could eliminate the abnormally accumulated HMGCR protein induced by various statins. In cultured cells, Cmpd 81 markedly decreased the HMGCR protein to an extremely low level in the presence or absence of lovastatin (Fig. 6a). Additionally, Cmpd 81 similarly reduced the amounts of HMGCR protein induced by the other five clinically prescribed statins: simvastatin, pravastatin,

| R | Cmpd | EC$_{50}$ (μM) | Cmpd | EC$_{50}$ (μM) | Cmpd | EC$_{50}$ (μM) | Cmpd | EC$_{50}$ (μM) |
|---|---|---|---|---|---|---|---|---|
| ⌇OH | 51 | 9.08 | 74 | 4.52 | 96 | 2.96 | 113 | 1.72 |
| ⌇O— | 55 | 2.29 | 78 | 1.85 | 95 | 2.34 | — | — |
| ⌇COOH | 50 | No activity | 73 | 60.57 | 92 | 12.34 | — | — |
| ⌇COO— | 49 | 6.67 | 72 | 6.77 | — | — | — | — |
| ⌇OH | 63 | 2.96 | 86 | 2.17 | 107 | 1.29 | 124 | 0.92 |
| ⌇OH | 64 | 3.06 | 87 | 1.08 | 108 | 1.03 | 125 | 1.83 |
| ⌇OH | 65 | 2.33 | 88 | 0.91 | 109 | 0.91 | 126 | 2.45 |
| ⌇OH | 56 | 52.10 | 79 | 0.64 | 100 | 0.51 | 117 | 0.71 |
| ⌇OH | 57 | 1.75 | 80 | 0.55 | 101 | 0.59 | 118 | 0.97 |
| ⌇OH | 58 | 4.50 | Cmpd 81 (HMG499) | 0.41 | 102 | 1.92 | 119 | 4.13 |

**Fig. 4** Summary of the EC$_{50}$ values of various compounds. Compounds bearing the 3β-hydroxyl group, 4,4-dimethyl groups and 7β-hydroxyl group on the steroid ring but differing in the side chain length or terminal moieties (R) were synthesized. CHG cells were incubated with indicated compounds for 16 h and the mean intensity of GFP from three independent experiments were measured. The EC$_{50}$ values were determined with dose–response curves by Prism software. Those with EC$_{50}$ values less than 0.75 μM are highlighted in shadow. Source data are provided as a Source Data File. Dose–response curves of indicated compounds are shown in Supplementary Fig. 3

fluvastatin, atorvastatin and rosuvastatin (Supplementary Fig. 6). To assess the in vivo effect, adult male mice on a chow diet were gavaged with vehicle, Cmpd 81, lovastatin, or both for 10 consecutive days, and the liver HMGCR protein levels were quantified after immunoblotting. While lovastatin elicited a more than 30-fold increase of HMGCR protein compared with vehicle, co-administration of Cmpd 81 led to a substantially decrease of HMGCR (>13-fold lower to lovastatin-treated samples and only ~2-fold to vehicle) (Fig. 6b). Besides, Cmpd 81 alone decreased the HMGCR amount by 50% (Fig. 6b).

We next placed mice on a medium-fat-medium-cholesterol diet (MFMC) and extended the treatments to 6 weeks, which is a relatively mild obesity-inducing diet mimicking the situation in humans better. Administration of Cmpd 81 or lovastatin alone significantly lowered total cholesterol (TC) and triglyceride (TG) levels in both serum and liver of mice fed on MFMC, and Cmpd 81 had comparable effects with lovastatin (Fig. 6c–f). Notably, addition of Cmpd 81 to lovastatin augmented its efficacy on reducing TC level in the serum and liver (Fig. 6c, e). Consistently, the serum lipoprotein profiles showed that cholesterol contents in very-low-density lipoprotein (VLDL), LDL and high-density lipoprotein (HDL) were all reduced to similar levels in Cmpd 81- and lovastatin-treated mice, and were all further lowered by the combination treatment of Cmpd 81 and lovastatin (Fig. 6g).

The body weight and food intake remained comparable in mice treated with Cmpd 81, lovastatin or both (Supplementary Fig. 7a,b).

We then accessed whether Cmpd 81 displayed any anti-atherogenic effects. The LDL receptor-deficient (Ldlr$^{-/-}$) mice on a western diet (WD), a well-established atherosclerosis model, were treated with vehicle, lovastatin, Cmpd 81 or both for 20 weeks. Mice receiving different treatments showed similar body weight and food intake (Supplementary Fig. 7c, d). However, single agent treatment of lovastatin and Cmpd 81 showed significant effects on alleviating the elevations of serum and liver TC and TG in Lldr$^{-/-}$ mice (Supplementary Fig. 7e–h). FPLC analysis revealed that Cmpd 81 and lovastatin remarkably reduced serum cholesterol levels in VLDL, LDL and HDL to similar contents, and Cmpd 81 synergized with lovastatin to further decrease cholesterol contents in all lipoproteins, especially VLDL and LDL (Fig. 6h). We next analyzed the atherosclerotic lesions in different groups using Sudan IV staining. Significantly less aorta lesions were observed in mice administered with lovastatin or Cmpd 81 compared with those with vehicle (Fig. 6i, j). Notably, the atherosclerotic plaques were further decreased by a combined treatment with lovastatin and Cmpd 81 (Fig. 6i, j). Together, these results suggest that compound Cmpd 81 can function alone or act synergistically

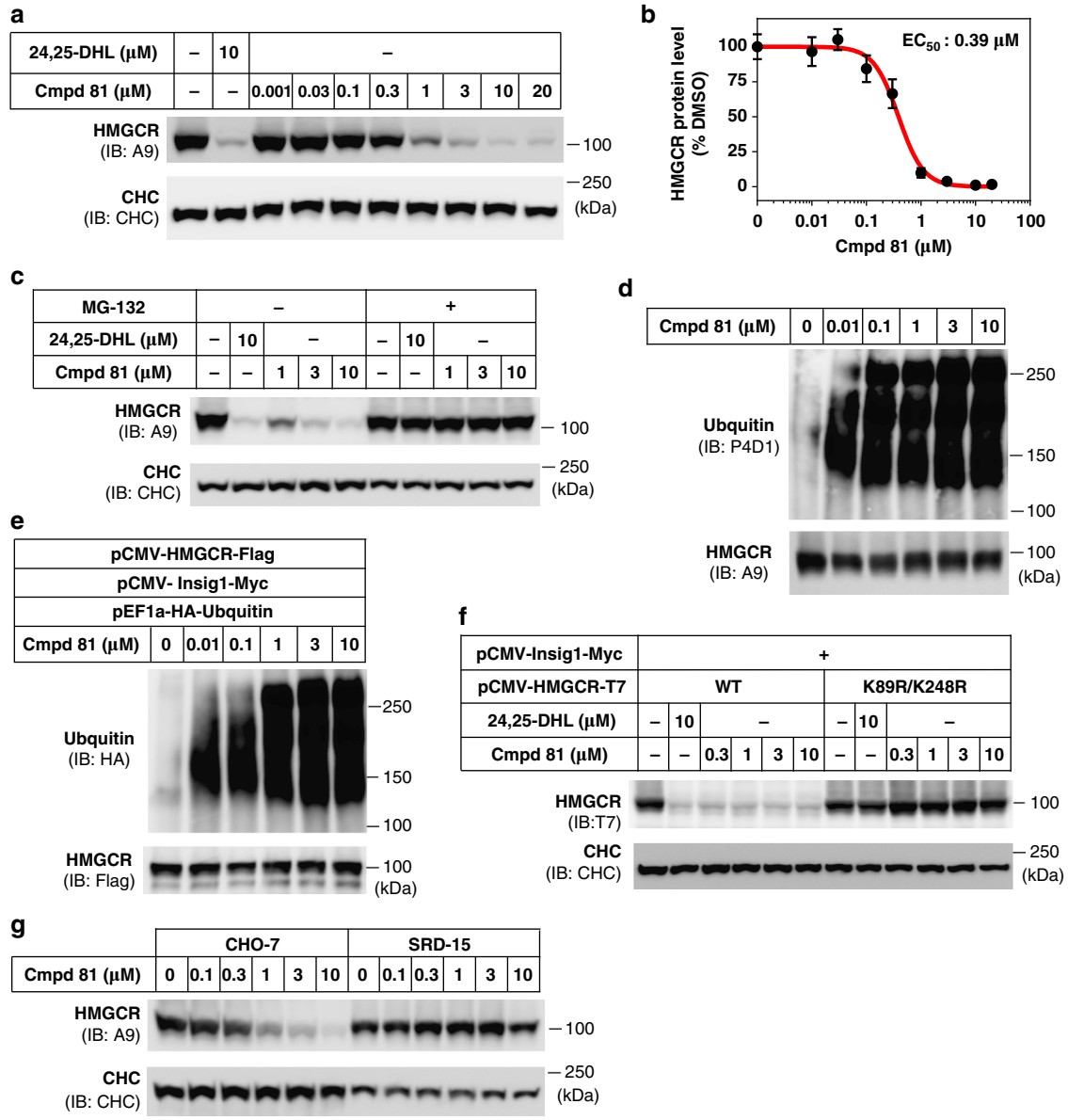

**Fig. 5** Cmpd 81 induces HMGCR ubiquitination and degradation. **a**, **b** CHO-7 cells were treated with Cmpd 81 at indicated concentrations for 16 h. 24,25-DHL was used as a positive control. **a** Immunoblotting analysis of HMGCR protein. **b** Quantification of Cmpd 81-induced HMGCR degradation shown in **a**. HMGCR protein of DMSO-treated cells was defined as 100. The $EC_{50}$ value of Cmpd 81 was 0.39 μM. **c** CHO-7 cells were treated with Cmpd 81 in the presence (+) or absence (−) of 10 μM MG-132 for 5 h. MG-132 prevented Cmpd 81-induced HMGCR degradation. **d** CHO-7 cells were treated with Cmpd 81 and 10 μM MG-132 for 3 h. Lysates were immunoprecipitated and pellets were probed for anti-ubiquitin (P4D1) and anti-HMGCR (A9). **e** CHO-7 cells were transfected with indicated plasmids and treated as in **d**. Lysates were immunoprecipitated with anti-Flag M2 beads, and pellets were probed for anti-ubiquitin (HA) and anti-HMGCR (Flag). **f** CHO-7 cells were transfected with indicated plasmids, then treated with Cmpd 81 for 5 h. Cmpd 81 promoted the degradation of WT but not ubiquitin sites mutated (K89R/K248R) HMGCR protein. **g** CHO-7 and SRD15 cells were treated with Cmpd 81 and the HMGCR protein was subjected for immunoblotting analysis. Data are from three independent experiments and presented as mean ± SD. Source data are provided as a Source Data File. Uncropped immunoblots are shown in Supplementary Fig. 10

## Discussion

Statins are the most prescribed drug regimen for treating CVDs since they were first introduced to the market in the late 1980s. An even broader population is eligible for statin therapy according to the 2013 ACC/AHA cholesterol guidelines[50–52]. Accompanying with the cardiovascular benefits, however, all kinds of statins block the generation of mevalonate derivatives that inhibit HMGCR expression through multiple feedback regulations, causing a remarkable increase of the enzyme (Fig. 1 and Supplementary Fig. 1). This, as a result, hampers the effectiveness of the drug, provoking more intensive treatments that elicit unfavorable side effects[25–29]. Our study provides a potential solution to this problem. In this study, we have successfully identified the cholesterol derivative Cmpd 81 that promotes the degradation of HMGCR without activating the lipogenic activity of LXR. Notably, Cmpd 81 profoundly reduces statin-induced increases of HMGCR protein in cultured cells and animals. It can

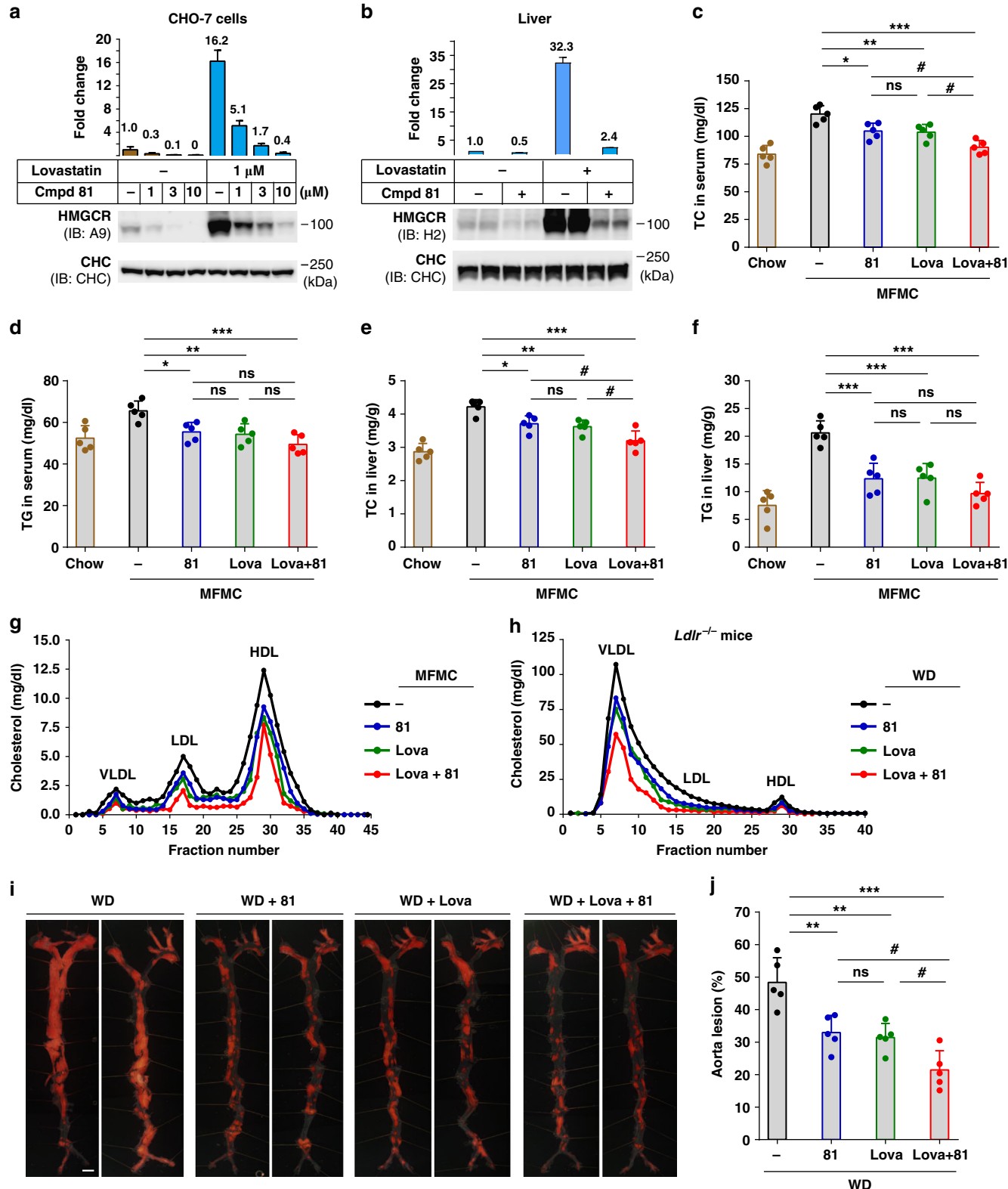

function alone as well as potentiate the efficacy of lovastatin in ameliorating cholesterol diet-induced hyperlipidemia and reducing atherosclerotic plaques. This study proves the concept that inducing HMGCR degradation can effectively reduce serum cholesterol levels and overcome the defect of statins (Fig. 7), highlighting a promising strategy for treating hypercholesterolemia and CVDs.

Recent studies indicate that hepatic accumulated free cholesterol is a critical lipotoxic molecule in the development of non-alcoholic steatohepatitis (NASH), as the increased hepatic cholesterol synthesis and expression of HMGCR are associated with the severity of NASH[53,54]. Moreover, statin usage was reported to reduce hepatic steatosis, inflammation and fibrosis in NASH patients[55]. Along the same vein, Cmpd 81 and lovastatin

**Fig. 6** Cmpd 81 prevents lovastatin-induced HMGCR accumulation and ameliorates diet-induced hyperlipidemia and atherosclerosis. **a** Immunoblotting and quantification of HMGCR protein from CHO-7 cells. Data are from three independent experiments. **b** Male C57BL/6J mice ($n = 3$ per group) fed on chow diet were gavaged once daily with Cmpd 81 (60 mg/kg/day) with or without lovastatin (60 mg/kg/day) for 10 days. HMGCR proteins from liver membrane fractions were analyzed and quantified by immunoblotting. **c–g** Male C57BL/6J mice ($n = 5$ per group) fed with chow diet or medium fat medium cholesterol (MFMC) diet, were gavaged once daily with Cmpd 81 (60 mg/kg/day) with or without lovastatin (60 mg/kg/day) for 6 weeks. Blood and livers were harvested and examined for total cholesterol (TC) levels in the serum (**c**), triglyceride (TG) levels in the serum (**d**), TC levels in liver (**e**), TG levels in liver (**f**), and lipoprotein profiles of cholesterol levels in pooled serum using FPLC analysis (**g**). **h–j** Male $Ldlr^{-/-}$ mice ($n = 5$ per group) fed on western diet (WD) were gavaged once daily with indicated Cmpd 81 (60 mg/kg/day) and lovastatin (60 mg/kg/day) for 20 weeks. **h** FPLC analysis of cholesterol contents in different lipoproteins. **i** Representative *en face* Sudan IV staining of aortas. Scale bar, 200 μm. **j** Quantifications of atherosclerotic lesions shown in **i**. Data are presented as mean ± SD. $P$ values were determined by one-way ANOVA followed by Dunnett's multiple comparisons test. $^{*}P <$ 0.05, $^{**}P < 0.01$, $^{***}P < 0.001$, $^{\#}P < 0.05$. Source data are provided as a Source Data File. Uncropped immunoblots are shown in Supplementary Fig. 10

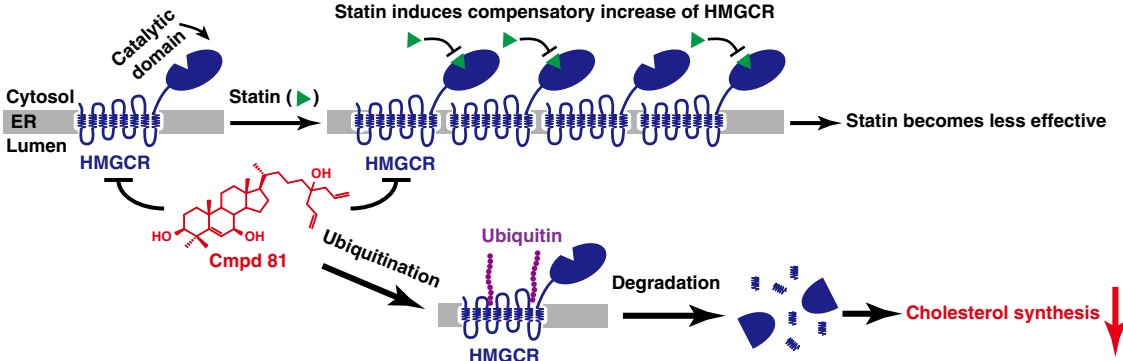

**Fig. 7** A working model of Cmpd 81 in lowering cholesterol synthesis. All clinically used statins dramatically induce compensatory increase of HMGCR, and the increased HMGCR proteins would blunt statins' efficacy. However, Cmpd 81 remarkably degrades statins-induced HMGCR protein through the ubiquitin-proteasome pathway to lower the cholesterol synthesis, and to improve the efficacy of statins

alone comparably reduce hepatic cholesterol synthesis in MFMC-fed or WD-fed $Ldlr^{-/-}$ mice, and Cmpd 81 synergizes with lovastatin to further decrease lipid levels (Fig. 6e and Supplementary Fig. 7g), indicating that induced degradation of HMGCR may bring therapeutic benefit to treat NASH. Thus, it will be interesting to evaluate the therapeutic effects of Cmpd 81 and statin, alone or synergistically, on the liver inflammation and fibrosis in NASH models. In addition, considering the pleiotropic effects of statin therapy related to the cholesterol lowering and mevalonate-derived isoprenoids depletion, it is reasonable to explore the combinatory effects of statin and Cmpd 81 in the conditions where statins alone are beneficial, such as cancer[56], central nervous system pathology[57], pulmonary alveolar proteinosis[58], vaccine adjuvant[59], and even cancer immunotherapy[59].

The design of compounds in our study was fueled by the knowledge of the molecular mechanisms of HMGCR degradation[13,35]. Cholesterol derivatives, including oxysterols (e.g., 25-HC, 27-HC) and methylated sterols (e.g., lanosterol and 24,25-DHL), can stimulate ubiquitination and proteasomal degradation of HMGCR[12,34–36,38,39,60]. We therefore synthesized series of cholesterol analogs and characterized their potency in stimulating HMGCR degradation using a GFP reporting system. We found that, in consistent with previous findings[12], dimethylation at C4 is important for sterol-regulated HMGCR degradation. Further, the potency is dramatically increased by adding a 7β-hydroxyl group and a 1,6-heptadien-4-hydroxyl group at the end of the side chain. Cmpd 81, one of the most effective chemicals, shows an $EC_{50}$ value approximately 74-fold and 4-fold more potent than that of cholesterol and 24,25-DHL, respectively. Further optimization on the chemical structure of Cmpd 81 may improve its activity and other drug-related properties.

Previous studies have suggested that 24,25-DHL triggers HMGCR degradation by directly targeting to the transmembrane domains (TM1-8) of HMGCR and stimulating its binding to the Insigs/gp78/VCP/Ufd1 complex followed by ubiquitination of the protein at lysine 89 and 248[12,34–36,60]. Following the same vain, substitution for these two lysine residues or ablation of $Insig$-$1/$-$2$ both completely abrogates the ability of Cmpd 81 to stimulate HMGCR degradation (Fig. 5f, g). Addition of the proteasome inhibitor MG-132 also blocks the HMGCR degradation triggered by Cmpd 81 (Fig. 5c). Considering the structurally similarity, it is very likely that Cmpd 81 and 24,25-DHL act to accelerate HMGCR degradation through the same mechanism, acting on the transmembrane domains (TM1-8) of HMGCR and requiring Insigs. It is still mysterious how the sterol-sensing domain of HMGCR binds and senses 24,25-DHL. It is necessary in the future to determine the structure of HMGCR(TM1-8) in both sterol-free and sterol-bound states. The structure information will provide critical insights into the mechanism of sterol-induced HMGCR degradation and guide for the design of more potent HMGCR degrader.

Originated from a natural cellular metabolite with a physiologically validated mechanism, Cmpd 81 (or this class of compounds) is certainly advantageous at least in two aspects. First, Cmpd 81 selectively eliminates HMGCR without up-regulating the expression of fatty acid biosynthetic master gene, $SREBP$-$1c$ (Supplementary Fig. 5), avoiding the side effects caused by oxysterols[46,47]. Secondly, unlike lanosterol and 24,25-DHL that are sterol intermediates of the cholesterol biosynthetic pathway, Cmpd 81 cannot be converted to cholesterol.

The safety profile of Cmpd 81 has also been evaluated. Administration of Cmpd 81 for up to 20 weeks in mice did not affect body weight and food intake. It appears to be safe and

efficacious to target HMGCR for degradation, and feasible for further drug development. Moreover, we evaluated the preclinical cardiac safety of Cmpd 81. The early preclinical cardiac safety assessment is a major concern in the drug discovery and development, given that many non-cardiovascular drugs (e.g., the gastric prokinetic drug cisapride) were withdrawn from the market because of the severe side effect of life-threatening cardiac arrhythmias[61]. Human *ether-a-go-go*-related gene (*HERG*) encodes heart highly expressed hERG potassium channels, and hERG current blockages are recognized as the predominant mechanism of drug-induced cardiac arrhythmias[62]. hERG electrophysiology patch-clamp assay showed that Cmpd 81 basically had no inhibition effect on hERG peak tail currents even at 40 µM, and the projected $IC_{50}$ of Cmpd 81 was as high as 1.86 mM, whereas the known hERG blocker cisapride strongly inhibited the current with an $IC_{50}$ of 0.031 µM (Supplementary Fig. 8). These data indicate that Cmpd 81 would be a good drug candidate without hERG channel associated cardiac toxicity.

Collectively, we have synthesized and characterized a series of compounds that effectively induces HMGCR degradation based on structure–activity relationship studies. Using Cmpd 81 as a representative, we prove the concept that this class of chemicals can prevent statins-induced accumulation of HMGCR, reduce serum cholesterol levels and decrease atherosclerosis. Our work suggests that inducing HMGCR degradation by Cmpd 81 or other chemicals can be a promising strategy alone or synergetic with statin therapy for the treatment of hyperlipidemia and atherosclerosis.

## Methods

**Reagents**. Cholesterol (purity ≥ 99%, C8667), T0901317 (T2320), sodium mevalonate (41288), sodium oleate (O3380), protease inhibitor cocktail (P8340), N-ethylmaleimide (E3876), paraformaldehyde (P6148), methyl cellulose (V900506), Tween-80 (P8074), crystal violet (229288) and Sudan IV (198102) were from Sigma-Aldrich; 24,25-dihydrolanosterol (purity ≥ 99%, 700067P) was from Avanti Lipids Polar; G418 (345810) was from Calbiochem; phenylmethylsulfonyl fluoride (52332) was from Merck; MG-132 (10012628) was from Cayman Chemical; leupeptin (11529048001) from was Roche Life Science; lovastatin (purity ≥ 98.5%, HPLC) was from Shanghai Pharm Valley, China; FuGENE HD transfection reagent (E2312) was from Promega; Hoechst 33342 (H1399) was from Invitrogen. Lipoprotein-deficient serum (LPDS)[63] and delipidated fetal calf serum[64] were prepared from fetal bovine serum (FBS) (S1580, Biowest) by ultracentrifugation in our laboratory.

**Antibodies**. Antibodies used for immunoblotting were as follows: mouse monoclonal anti-hamster HMGCR IgG-A9 (CRL-1811, ATCC, 2 µg ml$^{-1}$), mouse monoclonal anti-hamster SREBP-2 IgG-7D4 (CRL-2198, ATCC, 1 µg ml$^{-1}$), mouse monoclonal anti-clathrin heavy chain (610500, BD Biosciences, 1:1000), mouse monoclonal anti-ubiquitin antibody P4D1 (SC-8017, Santa Cruz Biotechnology, 1:1000), mouse monoclonal anti-HA (clone HA-7) (H3663, Sigma-Aldrich, 1:1000), mouse monoclonal anti-Flag (clone M2) (F3165, Sigma-Aldrich, 1:1000), mouse monoclonal anti-T7 (69522, Novagen, 1:5000). Rabbit polyclonal antibody H2 against HMGCR (1 µg ml$^{-1}$) and GFP (0.2 µg ml$^{-1}$) were prepared in our laboratory[65]. Horseradish peroxidase-conjugated goat anti-mouse (115-035-003, 1:5000) and anti-rabbit (111-035-144, 1:5000) secondary antibodies were from Jackson ImmunoResearch Laboratories.

**Plasmids**. pCMV-HMGCR (TM1-8)-GFP was constructed by inserting the nucleotide fragment, corresponding to the amino acids from 1 to 346 of hamster HMGCR, into pEGFP-N1 (Clontech), and confirmed by sequencing. pCMV-HMGCR-T7-WT, pCMV-HMGCR-T7-K89R/K248R, pCMV-HMGCR-Flag, pCMV-Insig1-Myc, pEF1a-HA-Ubiqintin were constructed in our laboratory. The primers used are listed in Supplementary Table 1.

**Media**. Medium A contains Dulbecco's modified Eagle's medium (DMEM) with 100 units ml$^{-1}$ penicillin and 100 µg ml$^{-1}$ streptomycin sulfate. Medium B contains a 1:1 mixture of DMEM and Ham's F-12 medium with 100 units ml$^{-1}$ penicillin and 100 µg ml$^{-1}$ streptomycin sulfate.

**Cell lines**. CHO-7 (Chinese hamster ovary cell, a clone of CHO-K1 cell line selected for growth in LPDS) and Huh7 (a human hepatocellular carcinoma cell line), SRD-13A (a *Scap*-deficient CHO-7 cell line) and SRD-15 (an *Insig-1* and

*Insig-2*-deficient CHO-7 cell line) were generous gifts from Russell Debose-Boyd at UT Southwestern Medical Center, USA. All cells were grown in a monolayer at 37 °C with 5% CO$_2$. Huh7 cells were maintained in medium A supplemented with 10% FBS. CHO-7 and SRD-15 cells were maintained in medium B supplemented with 5% FBS. SRD-13A cells were maintained in medium B with 5% FBS, 5 µg ml$^{-1}$ cholesterol, 1 mM sodium mevalonate, 10 µM sodium oleate. No cell lines used in this study were found in the database of commonly misidentified cell lines that is maintained by ICLAC and NCBI Biosample. The cell lines have not been authenticated recently. No test for mycoplasma contamination was performed.

**Generation of CHO-7/HMGCR (TM1-8)-GFP stable cell line**. CHO-7 cells were transfected with pCMV-HMGCR (TM1-8)-GFP using FuGENE HD. Twenty-four hours later, cells were switched to medium B supplemented with 5% FBS and selected in the presence of 1 mg ml$^{-1}$ G418 for 2 weeks. Surviving colonies were isolated and confirmed for GFP expression by immunoblotting and immunofluorescence. The resulting CHO-7/HMGCR (TM1-8)-GFP stable cell line (CHG) was maintained in medium B supplemented with 5% FBS and 200 µg ml$^{-1}$ G418.

**Quantification of HMGCR (TM1-8)-GFP intensity**. CHG cells were treated as described in figure legends. Briefly, cells were seeded at day 0 in 96-well black plate (3904, Corning) at a density of $1.5 \times 10^4$ cells per well, in medium B containing with 5% LPDS. Day 1, corresponding compounds were added as figure legends described in medium B with 5% LPDS, 1 µM lovastatin, 50 µM sodium mevalonate for 16 h. Then cells were fixed with 4% paraformaldehyde for 20 min, washed with PBS for three times, and stained with 2 µg ml$^{-1}$ Hoechst 33342.

The fluorescent images (GFP and Hoechst channel) were captured using an Operatta High-Content Imaging System (PerkinElmer) with a ×20 objective. The cytoplasmic GFP fluorescence was measured with the Harmony Software (PerkinElmer). Five random selected filed per well were captured, and about 3000 cells were used to determine the mean cytoplasmic GFP fluorescent intensity per cell after subtracting background GFP intensity. The average GFP intensity of DMSO-treated cells was defined as 100. The half-maximal effective concentration ($EC_{50}$) values were determined using nonlinear regression equation: log (inhibitor) vs response (variable slope, four parameters) by Prism 6 software (GraphPad).

**Immunoblotting**. Cells were harvested and homogenized in the RIPA buffer (50 mM Tris-HCl, pH 8.0, 150 mM NaCl, 2 mM MgCl$_2$, 1.5% NP-40, 0.1% SDS, and 0.5% sodium deoxycholate) supplemented with protease cocktail inhibitor, 10 µM MG-132, 10 µg ml$^{-1}$ leupeptin and 1 mM phenylmethylsulfonyl fluoride. The protein concentrations of lysates were determined using BCA method (Beyotime). Samples were mixed with the membrane protein solubilization buffer (62.5 mM Tris-HCl, pH 6.8, 15% SDS, 8 M urea, 10% glycerol, and 100 mM DTT) plus the 4× loading buffer (150 mM Tris-HCl, pH 6.8, 12% SDS, 30% glycerol, 6% 2-mercaptoethanol, and 0.02% bromophenol blue) and incubated for 30 min at 37 °C. Proteins were resolved by SDS-PAGE and transferred to nitrocellulose membranes (GE Healthcare). Immnoblots were blocked with 5% non-fat milk in TBS containing 0.075% Tween-20 (TBST) and probed with indicated antibodies overnight at 4 °C. After washing in TBST for five times, blots were incubated with HRP-conjugated secondary antibodies for 2 h at room temperature. Immunoreactivity was developed with Clarity Western ECL chemiluminescent substrates (170–5061, Bio-Rad). Western blot images were captured by Amersham Imager 600 (GE Healthcare), and integrated optical intensities of each band were quantified by Image-Pro Plus 6 software (Media Cybernetics). Uncropped blots are shown in Supplementary Figs. 9–13.

**Membrane fraction extraction**. Forty mg frozen liver was suspended in 800 µl ice-cold buffer A (10 mM HEPES/KOH, pH 7.6, 1.5 mM MgCl$_2$, 10 mM KCl, 5 mM EDTA, 5 mM EGTA, 250 mM sucrose) with protease inhibitors as described above, homogenized at 5000 r.p.m. 10 s with 3 times by Precellys 24 (Bertin), passed through #7 needle for 30 times, and then centrifuged at $1000 \times g$ for 10 min at 4 °C. The supernatant was removed to a new tube, and centrifuged at $1.6 \times 10^4 \times g$ for 15 min at 4 °C. The pellet was suspended in 0.1 ml SDS lysis buffer (10 mM Tris-HCl, pH 6.8, 100 mM NaCl, 1% (wt/vol) SDS, 1 mM EDTA, 1 mM EGTA) with protease inhibitors, and shaked at 1000 r.p.m. for 10 min at room temperature to completely dissolve the membrane proteins. The suspension was further centrifuged at $1.6 \times 10^4 \times g$ for 10 min at room temperature. A volume of 2 µl supernatant was taken for BCA protein quantification, and 100 µl supernatant was mixed with 100 µl above-described membrane protein solubilization buffer and 67 µl 4× loading buffer for 30 min at 37 °C, then for subsequent immunoblotting.

**Ubiquitination assays**. To detect endogenous HMGCR ubiquitination, CHO-7 cells were treated as described in figure legends. Two 10-cm dishes of cells were harvested and lysed in 600 µl of immunoprecipitation buffer (1× PBS, 1% Triton X-100, 5 mM EDTA, 5 mM EGTA) supplemented with protease inhibitor cocktail, 0.1 mM Leupeptin, 10 µM MG-132, and 10mM *N*-ethylmaleimide. Lysates were pre-cleared with 3 µg of normal rabbit IgG (D110502, Sangon) and 40 µl of protein A/G plus-agarose (sc-2003, Santa Cruz Biotechnology) for 1 h at 4 °C, followed by immunoprecipitation with 10 µg of rabbit polyclonal anti-HMGCR and 100 µl of protein A/G plus-agarose overnight at 4 °C. After washing with immunoprecipitation buffer containing protease

inhibitor cocktail for three times, immunoprecipitates were eluted with 2× loading buffer (75 mM Tris-HCl, pH 6.8, 50 mM NaCl, 6% SDS, 15% glycerol, and 0.02% bromophenol blue) at 95 °C for 10 min. Supernatants were then mixed with equal volume of membrane protein solubilization buffer and incubated at 37 °C for 30 min. Samples were finally subjected to immunoblotting as described above.

To assay for the ubiquitination of overexpressed HMGCR protein, cells were first transfected with indicated plasmids. A similar protocol was employed except that lysates were immunoprecipitated with 40 µl of anti-Flag M2 affinity gel (A2220, Sigma-Aldrich).

**Quantitative real-time PCR**. Cells were treated as described in corresponding figure legends, then total RNA was extracted from cells or mouse livers using TRI Reagent (T9424, Sigma-Aldrich). Equal amounts of RNA from cells or mice receiving the same treatment were pooled were used for cDNA synthesis with oligo dT and reverse transcriptase MLV (Promega). Gene expression was analyzed by quantitative real-time PCR on a Stratagene Mx30005PTM Q-PCR Systems. The primers used were listed in Supplementary Tables 2–4.

**Animals**. Male C57BL/6 mice (8-week-old) were obtained from Shanghai Laboratory Animal Company (SLAC) (Shanghai, China). Male $Ldlr^{-/-}$ mice (8-week-old) were obtained from Nanjing Biomedical Research Institute of Nanjing University (Nanjing, China). Mice had ad libitum access to water and standard chow diet (SLAC) unless mentioned otherwise. Special diets included medium-fat-medium-cholesterol diet (chow diet with10% fat, 0.2% cholesterol and 0.5% sodium cholate, custom made in SLAC), and western diet (20% fat, 1.25% cholesterol, 0.5% sodium cholate, D12109C, Research Diets). Mice were maintained in a pathogen-free environment and kept on a 12-h light/dark schedule. Mice were gavaged as described in figure legends with saline containing 0.5% methyl cellulose and 0.5% Tween-80. All mice were used in accordance with the guidelines approved by the Institutional Animal Care and Use Committee of Wuhan University.

**Serum and liver chemistry**. Mice were anesthetized and blood collection was performed. Serum was prepared from blood by centrifuging at $1500 \times g$ for 10 min. Livers were harvested and homogenized in a 2:1 mixture of chloroform and methanol using Precellys 24 (Bertin). The organic phase was dried under $N_2$ and liver lipids were reconstituted in ethanol. Total cholesterol and triglyceride levels in the serum and liver were determined using Cholesterol Assay Kit and Triglyceride Assay Kit (Shanghai Kehua Bio-engineering, China).

**Analysis of atherosclerotic lesions**. To examine aorta atherosclerotic lesions, the aortic tree was isolated and fixed with 4% PFA. After removing perivascular adipose tissues, the atherosclerotic lesions were stained with Sudan IV for 10 min and washed with 70% ethanol for 3 min. The aortic tree was pinned to the black wax-filled petri dish containing PBS and imaged under a stereoscopic microscope (Olympus SZX16). The atherosclerotic lesions were quantified with Image J software (NIH)[66].

**Fast protein liquid chromatography (FPLC)**. Serum lipoproteins were separated by FPLC (AKTA, GE Healthcare) using Superdex 75 10/300 GL (GE Healthcare) column. One hundred microliters pooled serum from five mice per group was diluted with 200 µl PBS, and loaded to the column. Fractions were eluted with PBS at a constant rate of 0.2 ml min$^{-1}$, and collected with 300 µl per tube. 100 µl fractions were used to quantify the concentration of total cholesterol level with the Amplex Red Cholesterol Assay Kit (Invitrogen). The cholesterol distribution curves of lipoproteins were done with GraphPad Prism 6 software.

**hERG electrophysiology patch-clamp assay**. CHO-hERG cells (Sophion Biosciences), which are CHO-K1 cells stably transfected with hERG, were detached with Detachin (AMS Biotechnology) solution and harvested as single cells suspension for the subsequent hERG electrophysiological assay. hERG whole-cell patch-clamp recording was conducted on a QPatch-16X automated electrophysiology platform (Sophion Biosciences) with 16-channel Qpatch plates (Sophion Biosciences). After whole-cell configuration was established, the voltage stimulation paradigm was set as followings: the cells were held at −80 mV, clamped at −50 mV for 50 ms to measure the leaking current, depolarized at +40 mV for 5 s to partly open potassium channels, repolarized at −50 mV for 5 s to fully open all channels and then to slowly close channels, finally held back to −80 mV. This paradigm was repeated once every 5 s to measure the current amplitude. The extracellular solution (control) was added firstly, cells were stimulated and measured with the above paradigm for 5 min. Then the tested compound was applied for 2.5 min from low concentrations to high concentrations gradually, at least 3 cells were measured for each concentration. The last three peak tail current values of each period treated with compounds were quantified with GraphPad Prism 6 software to determinate the IC$_{50}$ values. Cisapride (CDS021610, Sigma) was a positive hERG blocker.

**Statistical analyses**. All statistical analyses were performed using the GraphPad Prism 6 software. Data were expressed as means ± SD and analyzed by unpaired two-tailed Student's $t$-test or one-way ANOVA with Dunnett's multiple comparisons test as indicated. Statistical tests were justified as appropriate for every figure. Statistical significance was set at $P < 0.05$. Sample sizes, statistical tests and P values for each experiment are depicted in the relevant figure legends. Experiments on cultured cells were successfully repeated for three times. Experiments on mice were performed once with indicated $n$ and biological replicates.

**Chemical synthesis and characterization**. The synthesis procedure and spectra characterization of compounds used in this study are described in the Supplementary Figs. 14–139 and Supplementary methods.

**Reporting summary**. Further information on research design is available in the Nature Research Reporting Summary linked to this article.

## Data availability
The data that support the findings of this study are available within the paper and its supplementary information files, and from the corresponding author upon reasonable request.

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

## Acknowledgements

We thank Y.-X. Qu., B.-Y. Xiang. and J. Xu. for technical assistance and Zhao-Bing Gao (Shanghai Institute of Materia Medica, Chinese Academy of Sciences) for performing the hERG electrophysiology patch-clamp assay. This work was supported by grants from the NNSF of China (91753204, 31430044, and 31700711), MOST of China (2016YFA0500100), 111 Project of Ministry of Education of China (B16036), Shanghai Science and Technology Council (Grant 18ZR1411200) and The National Key Technology R&D Program (No. 2015BAK45B00).

## Author contributions

B.-L.S. conceived the project. W.-W.Q. and J.T. designed the chemical entities and their synthetic strategy. S.-Y.J., H.L., J.-J.T., J.W., J.L., J.-K.W., X.-J.S., W.Q., F.Y., W.-W.Q., and B.-L.S. designed the experiments. S.-Y.J., H.L., J.-J.T., J.W., J. L., B.L., J.-K.W., X.-J.S.,

and H.-W.C. conducted the experiments. S.-Y.J., H.L., J.L., W.Q., W.-W.Q. and B.-L.S. wrote the paper with the input from other authors.

## Additional information

**Competing interests:** The authors declare no competing interests.

