## [Peer Review File · Nature Communications]

Reviewers' comments:

Reviewer #1 (Remarks to the Author):

This manuscript by Jiang et al describes the discovery and optimization of degraders that lower the levels of HMGCoA reductase through degradation, thus counteracting the well-known statin-induced overexpression of this enzyme, eventually leading to a synergistic lowering of triglycerides in vivo. This is a very interesting finding that could have significant impact for further development. As such, I believe that a preliminary disclosure in Nature Communications after revision is justified.

Before going into the details of the review, I would have to add a disclaimer that I am a chemist and will focus my review (mostly) on the chemistry aspects of the work. I trust that the assay and in vivo work will be considered by experts in those areas.

The most glaring omission, which can presumably be fixed fairly easily, is that no information about the synthesis, characterization, or purity of the compounds studied is provided. This is essential information without which the work cannot be reliably evaluated or reproduced. Its inclusion in the Supporting Information is therefore absolutely essential. At the very least, the synthetic procedure, spectra, and purity measurements for the most important compounds (certainly compd 81 as well as any others that are important for the conclusions made in the manuscript such as 7, 35 or 79 and 80 as representatives of the series) needs to be provided. This should not be a problem since I am sure the authors have this data and hope that it is of (or can be brought to) publication quality.

The remaining issues concerns mostly wording and are relatively minor:

- Although the authors provide good evidence that the compounds are indeed degraders, this terms is nowadays often used in the context of PROTACs. In order to prevent confusion, it would be good to include a sentence early on making this distinction clear.
- Similarly, the term "rational optimizing" (sic) is usually reserved for structure- or ligand based design while the work here is more of a classical SAR study that is more of a trial-and-error than a rational design exercise.
- While the manuscript is overall well written, there is the occasional slip that needs to be fixed by another editorial pass. Besides the example above, other examples include "are little partly" "NH2-terminal" etc.
- On a broader level, the manuscript could be improved by streamlining the argument. The end of the introduction is really a summary of the work (and thus a repetition of the abstract) while the discussion is really more of a summary of the results rather than placing them in context or an exploration of how things might work.

Nevertheless, this is a nice piece of work that, with some additional data, is worthy of publication in Nature Communications.

Reviewer #2 (Remarks to the Author):

This work by Song and coworkers demonstrates that a novel cholesterol derivative, "Comp 81", is able to accelerate the degradation of HMG CoA reductase via ubiquitination enhancement. The effect of this compound is profound compared to previously known steroidal agents and has the additional attribute of being neutral with respect to potentially complicating counter targets associated with liver X receptor. Preliminary studies in mice are also encouraging.

The paper is concise and reasonably well-written (a copy edit should remove the few awkward terms and phrases in the paper). The figures are nicely done and the data appears to be of high quality.

I recommend the paper for acceptance with no substantial change except for the editorial modifications noted above.

Reviewer #3 (Remarks to the Author):

Statins are a widely prescribed class of drugs that decrease the production of cholesterol by inhibiting the rate limiting enzyme in cholesterol synthesis – HMG CoA reductase (HMGCR). Statins are effective and mostly well tolerated, but in some cases, there are side effects (e.g. myopathy). In addition, statins can cause an increase in HMGCR protein levels, necessitating increases in statin amounts to maintain efficacy. Reducing the levels of HMGCR is a strategy that could replace statins or facilitate the action of statins. Previous findings indicate that select cholesterol intermediates (e.g. lanosterol and 24,25-DHL) are potent inducers of HMGCR degradation. Unfortunately, lanosterol and 24,25-DHL induce fatty acid biosynthesis through LXR and SREBP-1c and they can also be converted to cholesterol.

In the current manuscript, the authors demonstrate that statin treatment increases HMGCR levels through a posttranscriptional mechanism, likely due to a decrease in ubiquitin-dependent HMGCR proteasomal clearance. Inspired by the effects of lanosterol and 24,25-DHL on HMGCR degradation, the authors examine a panel of synthetic cholesterol derivatives. Their findings identify several cholesterol derivatives (e.g. Cmpd 81) that potently induce HMGCR degradation via the proteasome. The induced degradation requires insigs and two key lysines that serve as sites for ubiquitination, suggesting that HMGCR is being degraded through the canonical ER-associated degradation (ERAD) pathways. Importantly, Cmpd 81 does not induce LXR / SREBP pathways and it cannot be converted to cholesterol. Furthermore, the authors find that Cmpd 81 is effective in the blocking statin-induced increases in HMGCR in cultured cells and in mice. Cmpd 81 – alone or in combination with lovastatin – is capable of decreasing liver triacylglycerol and cholesterol in mice fed a medium-fat-medium-cholesterol diet and decreasing aortic lesions in an atherosclerosis model.

Overall, the study is rigorously designed, the data presented are clear, and the conclusions are well supported. The identification of cholesterol-inspired molecules that potently induce HMGCR degradation and that demonstrate the therapeutic potential of this strategy is exciting and of broad interest.

Comments:

Comment 1: An important feature of these new compounds is that they do not induce LXR and SREBP pathways (Figure S4A), which would contrast with lanosterol and 24,25-DHL. It would be useful to compare the effects of lanosterol and 24,25-DHL with Cmpd 81 in Figure S4A, instead of the LXR agonist.

Comment 2: Please provide additional discussion of the drug-like properties of Cmpd 81. Is it possible to use this as a drug? Why or why not? Or is this simply a proof of concept for the strategy? Are there particular features that need to be optimized for delivery / efficacy?

We appreciate the constructive and insightful comments and suggestions from the reviewers. The specific points are addressed below.

Reviewer #1 (Remarks to the Author):

This manuscript by Jiang et al describes the discovery and optimization of degraders that lower the levels of HMGCoA reductase through degradation, thus counteracting the well-known statin-induced overexpression of this enzyme, eventually leading to a synergistic lowering of triglycerides in vivo. This is a very interesting finding that could have significant impact for further development. As such, I believe that a preliminary disclosure in Nature Communications after revision is justified.

Response: We thank the reviewer for his/her positive comments on our work.

Before going into the details of the review, I would have to add a disclaimer that I am a chemist and will focus my review (mostly) on the chemistry aspects of the work. I trust that the assay and in vivo work will be considered by experts in those areas.

The most glaring omission, which can presumably be fixed fairly easily, is that no information about the synthesis, characterization, or purity of the compounds studied is provided. This is essential information without which the work cannot be reliably evaluated or reproduced. Its inclusion in the Supporting Information is therefore absolutely essential. At the very least, the synthetic procedure, spectra, and purity measurements for the most important compounds (certainly compd 81 as well as any others that are important for the conclusions made in the manuscript such as 7, 35 or 79 and 80 as representatives of the series) needs to be provided. This should not be a problem since I am sure the authors have this data and hope that it is of (or can be brought to) publication quality.

Response: We appreciate the reviewer for pointing out this omission in our initial submission. We are sorry for omitting the “Supplementary Note 1” which contains synthetic procedures and characterization (¹H NMR, ¹³C NMR and HRMS) of all compounds used in this paper. Moreover, as suggested by the reviewer, the purity of compounds 7, 35, 79, 80 and 81 has been carried out using HPLC, and their purity are more than 95% at 210 nm and 205 nm. (details, please see the “Supplementary Note 1”).

The remaining issues concerns mostly wording and are relatively minor:

- Although the authors provide good evidence that the compounds are indeed degraders, this terms is nowadays often used in the context of PROTACs. In order to prevent confusion, it would be good to include a sentence early on making this distinction clear.

Response: We thank the reviewer for this helpful suggestion. We added a related sentence in introduction and cited the corresponding paper¹ (page 4).

- Similarly, the term “rational optimizing” (sic) is usually reserved for structure- or ligand based design while the work here is more of a classical SAR study that is more of a trial-and-error than a rational design exercise.

Response: We thank the reviewer for this helpful suggestion. We have replaced “rational optimizing” with “structure-activity relationship analysis”, and “rational design” with “structure-activity relationship studies” in the revised manuscript.

- While the manuscript is overall well written, there is the occasional slip that needs to be fixed by another editorial pass. Besides the example above, other examples include “are little partly” “NH2-terminal” etc.

Response: We thank the reviewer for pointing out these mistakes. We have replaced “are little partly” with “are little”, “NH2-terminal” with “N-terminal”, and “COOH-terminal” with “C-terminal”, and so on in the revised manuscript.

- On a broader level, the manuscript could be improved by streamlining the argument. The end of the introduction is really a summary of the work (and thus a repetition of the abstract) while the discussion is really more of a summary of the results rather than placing them in context or an exploration of how things might work.

Response: We thank the reviewer for these suggestions. We take the reviewer’s suggestion of simplifying the end of introduction and adding more discussions in the revised manuscript.

Nevertheless, this is a nice piece of work that, with some additional data, is worthy of publication in Nature Communications.

Response: We thank the reviewer for his/her positive comments on our work.

Reviewer #2 (Remarks to the Author):

This work by Song and coworkers demonstrates that a novel cholesterol derivative, “Comp 81”, is able to accelerate the degradation of HMG CoA reductase via ubiquitination enhancement. The effect of this compound is profound compared to previously known steroidal agents and has the additional attribute of being neutral with respect to potentially complicating counter targets associated with liver X receptor. Preliminary studies in mice are also encouraging.

The paper is concise and reasonably well-written (a copy edit should remove the few awkward terms and phrases in the paper). The figures are nicely done and the data appears to be of high quality.

I recommend the paper for acceptance with no substantial change except for the editorial modifications noted above.

Response: We thank the reviewer for his/her positive comments on our work, and suggestion on the editorial modifications. We have carefully checked and edited the English writing in our revised manuscript

Reviewer #3 (Remarks to the Author):

Statins are a widely prescribed class of drugs that decrease the production of cholesterol by inhibiting the rate limiting enzyme in cholesterol synthesis – HMG CoA reductase (HMGCR). Statins are effective and mostly well tolerated, but in some cases, there are side effects (e.g. myopathy). In addition, statins can cause an increase in HMGCR protein levels, necessitating increases in statin amounts to maintain efficacy. Reducing the levels of HMGCR is a strategy that could replace statins or facilitate the action of statins. Previous findings indicate that select cholesterol intermediates (e.g. lanosterol and 24,25-DHL) are potent inducers of HMGCR degradation. Unfortunately, lanosterol and 24,25-DHL induce fatty acid biosynthesis through LXR and SREBP-1c and they can also be converted to cholesterol.

In the current manuscript, the authors demonstrate that statin treatment increases HMGCR levels through a posttranscriptional mechanism, likely due to a decrease in ubiquitin-dependent HMGCR proteasomal clearance. Inspired by the effects of lanosterol and 24,25-DHL on HMGCR degradation, the authors examine a panel of synthetic cholesterol derivatives. Their findings identify several cholesterol derivatives (e.g. Cmpd 81) that potently induce HMGCR degradation via the proteasome. The induced degradation requires insigs and two key lysines that serve as sites for ubiquitination, suggesting that HMGCR is being degraded through the canonical ER-associated degradation (ERAD) pathways. Importantly, Cmpd 81 does not induce LXR / SREBP pathways and it cannot be converted to cholesterol. Furthermore, the authors find that Cmpd 81 is effective in the blocking statin-induced increases in HMGCR in cultured cells and in mice. Cmpd 81 – alone or in combination with lovastatin – is capable of decreasing liver triacylglycerol and cholesterol in mice fed a medium-fat-medium-cholesterol diet and decreasing aortic lesions in an atherosclerosis model.

Overall, the study is rigorously designed, the data presented are clear, and the conclusions are well supported. The identification of cholesterol-inspired molecules that potently induce HMGCR degradation and that demonstrate the therapeutic potential of this strategy is exciting and of broad interest.

Response: We thank the reviewer for his/her positive comments on our work and precisely summarizing our work.

Comments:

Comment 1: An important feature of these new compounds is that they do not induce LXR and SREBP pathways (Figure S4A), which would contrast with lanosterol and 24,25-DHL. It would be useful to compare the effects of lanosterol and 24,25-DHL with Cmpd 81 in Figure S4A, instead of the LXR agonist.

Response: We thank the reviewer for pointing out this possibility. Previous studies showed that lanosterol and 24,25-DHL had no effect on the activation of LXR². We here further validated the effects of lanosterol and 24,25-DHL on the expression of LXR target genes. **Figure for reviewers 1** shows that lanosterol and 24,25-DHL did not activate the LXR target genes such as: the genes related to fatty acid synthesis (*SREBP-1c*, *FASN* and *SCD1*) and cholesterol efflux genes (*ABCA1*, *ABCG5* and *ABCG8*), indicating that lanosterol and 24,25-DHL are not LXR agonists.

Figure for reviewers 1. Effects of lanosterol and 24,25-DHL on the expression of LXR target genes. Experiments were performed with the same condition as **Figure S4A**.

Comment 2: Please provide additional discussion of the drug-like properties of Cmpd 81. Is it possible to use this as a drug? Why or why not? Or is this simply a proof of concept for the strategy? Are there particular features that need to be optimized for delivery / efficacy?

Response: We thank the reviewer for this constructive suggestion. We took the reviewer's suggestions and added additional discussion of the drug-like properties of Cmpd 81 in the revised manuscript. Treatment of Cmpd 81 for up to 20 weeks in mice did not affect body weight and food intake, indicating that Cmpd 81 would be safe and feasible for further drug development.

Moreover, we evaluated the preclinical cardiac safety of Cmpd 81. The early preclinical cardiac safety assessment is a major concern in the drug discovery and development, given that many non-cardiovascular drugs (for example cisapride, a gastric prokinetic drug) were withdrawn from the market because of the severe side effect of life-threatening cardiac arrhythmias associated with the prolongation of QT interval³. Human *ether-a-go-go*-related gene (*HERG*) encodes heart highly expressed hERG potassium channels, and hERG current blockages are recognized as the predominant mechanism of drug-induced cardiac arrhythmias⁴. Therefore, we used an automated patch-clamp platform to evaluate the effect of Cmpd 81 on the hERG current. **Supplementary Figure 7** shows that the known hERG blocker cisapride strongly inhibited the peak tail current of hERG with an IC₅₀ of 0.031 μM. However, Cmpd 81 basically had no inhibition effect on hERG current even at 40 μM, and the projected IC₅₀ of Cmpd 81 was as high as 1.86 mM. The EC₅₀ of Cmpd 81 on HMGCR degradation is 0.39 μM (Figure 4b), which is 4769-fold lower. These data indicate that Cmpd 81 would be a good drug candidate without hERG potassium channel associated cardiac toxicity.

Comment 2: Are there particular features that need to be optimized for delivery / efficacy?

Response: Cmpd 81 seems to be lipophilic, like other sterols. This property may affect its absorption and distribution. We will try to reduce the lipophilicity of Cmpd 81 through further chemical structure optimization. As the reviewer pointed out, this work is a proof-of-concept study for HMGCR degraders. In the future, more efforts are needed to take to push the compound to clinical test.

Reference List

1. Lai, A.C. & Crews, C.M. Induced protein degradation: an emerging drug discovery paradigm. *Nat. Rev. Drug Discov.* **16**, 101-114 (2017).
2. Yang, C. *et al.* Sterol intermediates from cholesterol biosynthetic pathway as liver X receptor ligands. *J. Biol. Chem.* **281**, 27816-27826 (2006).
3. Fermini, B. & Fossa, A.A. The impact of drug-induced QT interval prolongation on drug discovery and development. *Nat. Rev. Drug Discov.* **2**, 439-447 (2003).
4. Sanguinetti, M.C. & Tristani-Firouzi, M. hERG potassium channels and cardiac arrhythmia. *Nature* **440**, 463-469 (2006).

REVIEWERS' COMMENTS:

Reviewer #1 (Remarks to the Author):

The authors have included the requested data, which are of publication quality. In addition, they have made several editorial changes to address my earlier points. The manuscript in the present form is recommended for publication

Reviewer #3 (Remarks to the Author):

The authors have addressed my comments and I recommend acceptance / publication. Very interesting and well executed study.

REVIEWERS' COMMENTS:

Reviewer #1 (Remarks to the Author):

The authors have included the requested data, which are of publication quality. In addition, they have made several editorial changes to address my earlier points. The manuscript in the present form is recommended for publication

Response: We thank the reviewer for his/her positive comments on our work and the recommendation for publication.

Reviewer #3 (Remarks to the Author):

The authors have addressed my comments and I recommend acceptance / publication. Very interesting and well executed study.

Response: We thank the reviewer for his/her positive comments on our work and the recommendation for publication.